# Bridging the Gap Between Value and Policy Based Reinforcement Learning

**Ofir Nachum**[1]     **Mohammad Norouzi**     **Kelvin Xu**[1]     **Dale Schuurmans**

`{ofirnachum,mnorouzi,kelvinxx}@google.com`, `daes@ualberta.ca`

Google Brain

## Abstract

We establish a new connection between value and policy based reinforcement learning (RL) based on a relationship between softmax temporal value consistency and policy optimality under entropy regularization. Specifically, we show that softmax consistent action values correspond to optimal entropy regularized policy probabilities *along any action sequence*, regardless of provenance. From this observation, we develop a new RL algorithm, *Path Consistency Learning (PCL)*, that minimizes a notion of soft consistency error along multi-step action sequences extracted from both on- and off-policy traces. We examine the behavior of PCL in different scenarios and show that PCL can be interpreted as generalizing both actor-critic and Q-learning algorithms. We subsequently deepen the relationship by showing how a *single* model can be used to represent both a policy and the corresponding softmax state values, eliminating the need for a separate critic. The experimental evaluation demonstrates that PCL significantly outperforms strong actor-critic and Q-learning baselines across several benchmarks.[2]

## 1 Introduction

Model-free RL aims to acquire an effective behavior policy through trial and error interaction with a black box environment. The goal is to optimize the quality of an agent's behavior policy in terms of the total expected discounted reward. Model-free RL has a myriad of applications in games [22, 37], robotics [16, 17], and marketing [18, 38], to name a few. Recently, the impact of model-free RL has been expanded through the use of deep neural networks, which promise to replace manual feature engineering with end-to-end learning of value and policy representations. Unfortunately, a key challenge remains how best to combine the advantages of value and policy based RL approaches in the presence of deep function approximators, while mitigating their shortcomings. Although recent progress has been made in combining value and policy based methods, this issue is not yet settled, and the intricacies of each perspective are exacerbated by deep models.

The primary advantage of policy based approaches, such as REINFORCE [45], is that they directly optimize the quantity of interest while remaining stable under function approximation (given a sufficiently small learning rate). Their biggest drawback is sample inefficiency: since policy gradients are estimated from rollouts the variance is often extreme. Although policy updates can be improved by the use of appropriate geometry [14, 27, 32], the need for variance reduction remains paramount. Actor-critic methods have thus become popular [33, 34, 36], because they use value approximators to replace rollout estimates and reduce variance, at the cost of some bias. Nevertheless, on-policy learning remains inherently sample inefficient [10]; by estimating quantities defined by the *current* policy, either on-policy data must be used, or updating must be sufficiently slow to avoid significant bias. Naive importance correction is hardly able to overcome these shortcomings in practice [28, 29].

By contrast, value based methods, such as Q-learning [44, 22, 30, 42, 21], can learn from *any* trajectory sampled from the same environment. Such "off-policy" methods are able to exploit data from other sources, such as experts, making them inherently more sample efficient than on-policy methods [10]. Their key drawback is that off-policy learning does not stably interact with function approximation [35, Chap.11]. The practical consequence is that extensive hyperparameter tuning can be required to obtain stable behavior. Despite practical success [22], there is also little theoretical understanding of how deep Q-learning might obtain near-optimal objective values.

Ideally, one would like to combine the unbiasedness and stability of on-policy training with the data efficiency of off-policy approaches. This desire has motivated substantial recent work on *off-policy* actor-critic methods, where the data efficiency of policy gradient is improved by training an off-policy critic [19, 21, 10]. Although such methods have demonstrated improvements over on-policy actor-critic approaches, they have not resolved the theoretical difficulty associated with off-policy learning under function approximation. Hence, current methods remain potentially unstable and require specialized algorithmic and theoretical development as well as delicate tuning to be effective in practice [10, 41, 8].

In this paper, we exploit a relationship between policy optimization under entropy regularization and softmax value consistency to obtain a new form of stable off-policy learning. Even though entropy regularized policy optimization is a well studied topic in RL [46, 39, 40, 47, 5, 4, 6, 7]–in fact, one that has been attracting renewed interest from concurrent work [25, 11]–we contribute new observations to this study that are essential for the methods we propose: first, we identify a strong form of path consistency that relates optimal policy probabilities under entropy regularization to softmax consistent state values for *any* action sequence; second, we use this result to formulate a novel optimization objective that allows for a stable form of off-policy actor-critic learning; finally, we observe that under this objective the actor and critic can be unified in a single model that coherently fulfills both roles.

## 2   Notation & Background

We model an agent's behavior by a parametric distribution $\pi_\theta(a \mid s)$ defined by a neural network over a finite set of actions. At iteration $t$, the agent encounters a state $s_t$ and performs an action $a_t$ sampled from $\pi_\theta(a \mid s_t)$. The environment then returns a scalar reward $r_t$ and transitions to the next state $s_{t+1}$.

**Note**: Our main results identify specific properties that hold for arbitrary action sequences. To keep the presentation clear and focus attention on the key properties, we provide a simplified presentation in the main body of this paper by assuming *deterministic* state dynamics. This restriction is not necessary, and in the Supplementary Material we provide a full treatment of the same concepts generalized to stochastic state dynamics. All of the desired properties continue to hold in the general case and the algorithms proposed remain unaffected.

For simplicity, we assume the per-step reward $r_t$ and the next state $s_{t+1}$ are given by functions $r_t = r(s_t, a_t)$ and $s_{t+1} = f(s_t, a_t)$ specified by the environment. We begin the formulation by reviewing the key elements of Q-learning [43, 44], which uses a notion of hard-max Bellman backup to enable off-policy TD control. First, observe that the expected discounted reward objective, $O_{\mathrm{ER}}(s, \pi)$, can be recursively expressed as,

$$O_{\mathrm{ER}}(s, \pi) \;=\; \sum_a \pi(a \mid s)\left[r(s, a) + \gamma O_{\mathrm{ER}}(s', \pi)\right], \qquad \text{where } s' = f(s, a) \,. \tag{1}$$

Let $V^\circ(s)$ denote the optimal state value at a state $s$ given by the maximum value of $O_{\mathrm{ER}}(s, \pi)$ over policies, *i.e.,* $V^\circ(s) = \max_\pi O_{\mathrm{ER}}(s, \pi)$. Accordingly, let $\pi^\circ$ denote the optimal policy that results in $V^\circ(s)$ (for simplicity, assume there is one unique optimal policy), *i.e.,* $\pi^\circ = \mathrm{argmax}_\pi O_{\mathrm{ER}}(s, \pi)$. Such an optimal policy is a one-hot distribution that assigns a probability of 1 to an action with maximal return and 0 elsewhere. Thus we have

$$V^\circ(s) \;=\; O_{\mathrm{ER}}(s, \pi^\circ) = \max_a (r(s, a) + \gamma V^\circ(s')). \tag{2}$$

This is the well-known hard-max Bellman temporal consistency. Instead of state values, one can equivalently (and more commonly) express this consistency in terms of optimal action values, $Q^\circ$:

$$Q^\circ(s, a) \;=\; r(s, a) + \gamma \max_{a'} Q^\circ(s', a') \,. \tag{3}$$

Q-learning relies on a value iteration algorithm based on (3), where $Q(s, a)$ is bootstrapped based on successor action values $Q(s', a')$.

## 3 Softmax Temporal Consistency

In this paper, we study the optimal state and action values for a *softmax* form of temporal consistency [48, 47, 7], which arises by augmenting the standard expected reward objective with a *discounted entropy regularizer*. Entropy regularization [46] encourages exploration and helps prevent early convergence to sub-optimal policies, as has been confirmed in practice (*e.g.,* [21, 24]). In this case, one can express regularized expected reward as a sum of the expected reward and a discounted entropy term,

$$O_{\text{ENT}}(s, \pi) = O_{\text{ER}}(s, \pi) + \tau \mathbb{H}(s, \pi) , \tag{4}$$

where $\tau \geq 0$ is a user-specified temperature parameter that controls the degree of entropy regularization, and the discounted entropy $\mathbb{H}(s, \pi)$ is recursively defined as

$$\mathbb{H}(s, \pi) = \sum_a \pi(a \mid s) \left[ -\log \pi(a \mid s) + \gamma \mathbb{H}(s', \pi) \right] . \tag{5}$$

The objective $O_{\text{ENT}}(s, \pi)$ can then be re-expressed recursively as,

$$O_{\text{ENT}}(s, \pi) = \sum_a \pi(a \mid s) \left[ r(s, a) - \tau \log \pi(a \mid s) + \gamma O_{\text{ENT}}(s', \pi) \right] . \tag{6}$$

Note that when $\gamma = 1$ this is equivalent to the entropy regularized objective proposed in [46].

Let $V^*(s) = \max_\pi O_{\text{ENT}}(s, \pi)$ denote the soft optimal state value at a state $s$ and let $\pi^*(a \mid s)$ denote the optimal policy at $s$ that attains the maximum of $O_{\text{ENT}}(s, \pi)$. When $\tau > 0$, the optimal policy is no longer a one-hot distribution, since the entropy term prefers the use of policies with more uncertainty. We characterize the optimal policy $\pi^*(a \mid s)$ in terms of the $O_{\text{ENT}}$-optimal state values of successor states $V^*(s')$ as a Boltzmann distribution of the form,

$$\pi^*(a \mid s) \propto \exp\{(r(s, a) + \gamma V^*(s'))/\tau\} . \tag{7}$$

It can be verified that this is the solution by noting that the $O_{\text{ENT}}(s, \pi)$ objective is simply a $\tau$-scaled constant-shifted KL-divergence between $\pi$ and $\pi^*$, hence the optimum is achieved when $\pi = \pi^*$.

To derive $V^*(s)$ in terms of $V^*(s')$, the policy $\pi^*(a \mid s)$ can be substituted into (6), which after some manipulation yields the intuitive definition of optimal state value in terms of a softmax (*i.e.,* log-sum-exp) backup,

$$V^*(s) = O_{\text{ENT}}(s, \pi^*) = \tau \log \sum_a \exp\{(r(s, a) + \gamma V^*(s'))/\tau\} . \tag{8}$$

Note that in the $\tau \to 0$ limit one recovers the hard-max state values defined in (2). Therefore we can equivalently state softmax temporal consistency in terms of optimal action values $Q^*(s, a)$ as,

$$Q^*(s, a) = r(s, a) + \gamma V^*(s') = r(s, a) + \gamma \tau \log \sum_{a'} \exp(Q^*(s', a')/\tau) . \tag{9}$$

Now, much like Q-learning, the consistency equation (9) can be used to perform one-step backups to asynchronously bootstrap $Q^*(s, a)$ based on $Q^*(s', a')$. In the Supplementary Material we prove that such a procedure, in the tabular case, converges to a unique fixed point representing the optimal values.

We point out that the notion of softmax Q-values has been studied in previous work (*e.g.,* [47, 48, 13, 5, 3, 7]). Concurrently to our work, [11] has also proposed a soft Q-learning algorithm for continuous control that is based on a similar notion of softmax temporal consistency. However, we contribute new observations below that lead to the novel training principles we explore.

## 4 Consistency Between Optimal Value & Policy

We now describe the main technical contributions of this paper, which lead to the development of two novel off-policy RL algorithms in Section 5. The first key observation is that, for the softmax

value function $V^*$ in (8), the quantity $\exp\{V^*(s)/\tau\}$ also serves as the normalization factor of the optimal policy $\pi^*(a \mid s)$ in (7); that is,

$$\pi^*(a \mid s) \;=\; \frac{\exp\{(r(s,a) + \gamma V^*(s'))/\tau\}}{\exp\{V^*(s)/\tau\}}\;. \tag{10}$$

Manipulation of (10) by taking the log of both sides then reveals an important connection between the optimal state value $V^*(s)$, the value $V^*(s')$ of the successor state $s'$ reached from *any* action $a$ taken in $s$, and the corresponding action probability under the optimal log-policy, $\log \pi^*(a \mid s)$.

**Theorem 1.** *For $\tau > 0$, the policy $\pi^*$ that maximizes $O_{ENT}$ and state values $V^*(s) = max_\pi O_{ENT}(s, \pi)$ satisfy the following temporal consistency property for any state $s$ and action $a$ (where $s' = f(s, a)$),*

$$V^*(s) - \gamma V^*(s') \;=\; r(s,a) - \tau \log \pi^*(a \mid s)\;. \tag{11}$$

*Proof. All theorems are established for the general case of a stochastic environment and discounted infinite horizon problems in the Supplementary Material. Theorem 1 follows as a special case.* □

Note that one can also characterize $\pi^*$ in terms of $Q^*$ as

$$\pi^*(a \mid s) \;=\; \exp\{(Q^*(s,a) - V^*(s))/\tau\}\;. \tag{12}$$

An important property of the one-step softmax consistency established in (11) is that it can be extended to a *multi-step* consistency defined on any action *sequence* from any given state. That is, the softmax optimal state values at the beginning and end of any action sequence can be related to the rewards and optimal log-probabilities observed along the trajectory.

**Corollary 2.** *For $\tau > 0$, the optimal policy $\pi^*$ and optimal state values $V^*$ satisfy the following extended temporal consistency property, for any state $s_1$ and any action sequence $a_1, ..., a_{t-1}$ (where $s_{i+1} = f(s_i, a_i)$):*

$$V^*(s_1) - \gamma^{t-1} V^*(s_t) \;=\; \sum_{i=1}^{t-1} \gamma^{i-1}[r(s_i, a_i) - \tau \log \pi^*(a_i \mid s_i)]\;. \tag{13}$$

*Proof. The proof in the Supplementary Material applies (the generalized version of) Theorem 1 to any $s_1$ and sequence $a_1, ..., a_{t-1}$, summing the left and right hand sides of (the generalized version of) (11) to induce telescopic cancellation of intermediate state values. Corollary 2 follows as a special case.* □

Importantly, the converse of Theorem 1 (and Corollary 2) also holds:

**Theorem 3.** *If a policy $\pi(a \mid s)$ and state value function $V(s)$ satisfy the consistency property (11) for all states $s$ and actions $a$ (where $s' = f(s, a)$), then $\pi = \pi^*$ and $V = V^*$. (See the Supplementary Material.)*

Theorem 3 motivates the use of one-step and multi-step path-wise consistencies as the foundation of RL algorithms that aim to learn parameterized policy and value estimates by minimizing the discrepancy between the left and right hand sides of (11) and (13).

## 5 Path Consistency Learning (PCL)

The temporal consistency properties between the optimal policy and optimal state values developed above lead to a natural path-wise objective for training a policy $\pi_\theta$, parameterized by $\theta$, and a state value function $V_\phi$, parameterized by $\phi$, via the minimization of a soft consistency error. Based on (13), we first define a notion of soft consistency for a $d$-length sub-trajectory $s_{i:i+d} \equiv (s_i, a_i, \ldots, s_{i+d-1}, a_{i+d-1}, s_{i+d})$ as a function of $\theta$ and $\phi$:

$$C(s_{i:i+d}, \theta, \phi) = -V_\phi(s_i) + \gamma^d V_\phi(s_{i+d}) + \sum_{j=0}^{d-1} \gamma^j [r(s_{i+j}, a_{i+j}) - \tau \log \pi_\theta(a_{i+j} \mid s_{i+j})]\;. \tag{14}$$

The goal of a learning algorithm can then be to find $V_\phi$ and $\pi_\theta$ such that $C(s_{i:i+d}, \theta, \phi)$ is as close to 0 as possible for all sub-trajectories $s_{i:i+d}$. Accordingly, we propose a new learning algorithm, called

*Path Consistency Learning (PCL)*, that attempts to minimize the squared soft consistency error over a set of sub-trajectories $E$,

$$O_{\text{PCL}}(\theta, \phi) = \sum_{s_{i:i+d} \in E} \frac{1}{2} C(s_{i:i+d}, \theta, \phi)^2. \tag{15}$$

The PCL update rules for $\theta$ and $\phi$ are derived by calculating the gradient of (15). For a given trajectory $s_{i:i+d}$ these take the form,

$$\Delta\theta = \eta_\pi C(s_{i:i+d}, \theta, \phi) \sum_{j=0}^{d-1} \gamma^j \nabla_\theta \log \pi_\theta(a_{i+j} \mid s_{i+j}), \tag{16}$$

$$\Delta\phi = \eta_v C(s_{i:i+d}, \theta, \phi) \left( \nabla_\phi V_\phi(s_i) - \gamma^d \nabla_\phi V_\phi(s_{i+d}) \right), \tag{17}$$

where $\eta_v$ and $\eta_\pi$ denote the value and policy learning rates respectively. Given that the consistency property must hold on *any* path, the PCL algorithm applies the updates (16) and (17) both to trajectories sampled on-policy from $\pi_\theta$ as well as trajectories sampled from a replay buffer. The union of these trajectories comprise the set $E$ used in (15) to define $O_{\text{PCL}}$.

Specifically, given a fixed rollout parameter $d$, at each iteration, PCL samples a batch of on-policy trajectories and computes the corresponding parameter updates for each sub-trajectory of length $d$. Then PCL exploits off-policy trajectories by maintaining a replay buffer and applying additional updates based on a batch of episodes sampled from the buffer at each iteration. We have found it beneficial to sample replay episodes proportionally to exponentiated reward, mixed with a uniform distribution, although we did not exhaustively experiment with this sampling procedure. In particular, we sample a full episode $s_{0:T}$ from the replay buffer of size $B$ with probability $0.1/B + 0.9 \cdot \exp(\alpha \sum_{i=0}^{T-1} r(s_i, a_i))/Z$, where we use no discounting on the sum of rewards, $Z$ is a normalization factor, and $\alpha$ is a hyper-parameter. Pseudocode of PCL is provided in the Appendix.

We note that in stochastic settings, our squared inconsistency objective approximated by Monte Carlo samples is a biased estimate of the true squared inconsistency (in which an expectation over stochastic dynamics occurs inside rather than outside the square). This issue arises in Q-learning as well, and others have proposed possible remedies which can also be applied to PCL [2].

## 5.1   Unified Path Consistency Learning (Unified PCL)

The PCL algorithm maintains a separate model for the policy and the state value approximation. However, given the soft consistency between the state and action value functions (*e.g.,*in (9)), one can express the soft consistency errors strictly in terms of Q-values. Let $Q_\rho$ denote a model of action values parameterized by $\rho$, based on which one can estimate both the state values and the policy as,

$$V_\rho(s) = \tau \log \sum_a \exp\{Q_\rho(s, a)/\tau\}, \tag{18}$$

$$\pi_\rho(a \mid s) = \exp\{(Q_\rho(s, a) - V_\rho(s))/\tau\}. \tag{19}$$

Given this unified parameterization of policy and value, we can formulate an alternative algorithm, called *Unified Path Consistency Learning (Unified PCL)*, which optimizes the same objective (*i.e.,* (15)) as PCL but differs by combining the policy and value function into a single model. Merging the policy and value function models in this way is significant because it presents a new actor-critic paradigm where the policy (actor) is not distinct from the values (critic). We note that in practice, we have found it beneficial to apply updates to $\rho$ from $V_\rho$ and $\pi_\rho$ using different learning rates, very much like PCL. Accordingly, the update rule for $\rho$ takes the form,

$$\Delta\rho = \eta_\pi C(s_{i:i+d}, \rho) \sum_{j=0}^{d-1} \gamma^j \nabla_\rho \log \pi_\rho(a_{i+j} \mid s_{i+j}) + \tag{20}$$

$$\eta_v C(s_{i:i+d}, \rho) \left( \nabla_\rho V_\rho(s_i) - \gamma^d \nabla_\rho V_\rho(s_{i+d}) \right). \tag{21}$$

## 5.2   Connections to Actor-Critic and Q-learning

To those familiar with advantage-actor-critic methods [21] (A2C and its asynchronous analogue A3C) PCL's update rules might appear to be similar. In particular, advantage-actor-critic is an on-policy method that exploits the expected value function,

$$V^\pi(s) = \sum_a \pi(a \mid s) \left[ r(s, a) + \gamma V^\pi(s') \right], \tag{22}$$

to reduce the variance of policy gradient, in service of maximizing the expected reward. As in PCL, two models are trained concurrently: an actor $\pi_\theta$ that determines the policy, and a critic $V_\phi$ that is trained to estimate $V^{\pi_\theta}$. A fixed rollout parameter $d$ is chosen, and the advantage of an on-policy trajectory $s_{i:i+d}$ is estimated by

$$A(s_{i:i+d}, \phi) = -V_\phi(s_i) + \gamma^d V_\phi(s_{i+d}) + \sum_{j=0}^{d-1} \gamma^j r(s_{i+j}, a_{i+j}) . \tag{23}$$

The advantage-actor-critic updates for $\theta$ and $\phi$ can then be written as,

$$\Delta\theta = \eta_\pi \mathbb{E}_{s_{i:i+d}|\theta} \left[ A(s_{i:i+d}, \phi) \nabla_\theta \log \pi_\theta(a_i|s_i) \right] , \tag{24}$$

$$\Delta\phi = \eta_v \mathbb{E}_{s_{i:i+d}|\theta} \left[ A(s_{i:i+d}, \phi) \nabla_\phi V_\phi(s_i) \right] , \tag{25}$$

where the expectation $\mathbb{E}_{s_{i:i+d}|\theta}$ denotes sampling from the current policy $\pi_\theta$. These updates exhibit a striking similarity to the updates expressed in (16) and (17). In fact, if one takes PCL with $\tau \to 0$ and omits the replay buffer, a slight variation of A2C is recovered. In this sense, one can interpret PCL as a generalization of A2C. Moreover, while A2C is restricted to on-policy samples, PCL minimizes an inconsistency measure that is defined on any path, hence it can exploit replay data to enhance its efficiency via off-policy learning.

It is also important to note that for A2C, it is essential that $V_\phi$ tracks the non-stationary target $V^{\pi_\theta}$ to ensure suitable variance reduction. In PCL, no such tracking is required. This difference is more dramatic in Unified PCL, where a single model is trained both as an actor and a critic. That is, it is not necessary to have a separate actor and critic; the actor itself can serve as its own critic.

One can also compare PCL to hard-max temporal consistency RL algorithms, such as Q-learning [43]. In fact, setting the rollout to $d = 1$ in Unified PCL leads to a form of soft Q-learning, with the degree of softness determined by $\tau$. We therefore conclude that the path consistency-based algorithms developed in this paper also generalize Q-learning. Importantly, PCL and Unified PCL are not restricted to single step consistencies, which is a major limitation of Q-learning. While some have proposed using multi-step backups for hard-max Q-learning [26, 21], such an approach is not theoretically sound, since the rewards received after a non-optimal action do not relate to the hard-max Q-values $Q^\circ$. Therefore, one can interpret the notion of temporal consistency proposed in this paper as a sound generalization of the one-step temporal consistency given by hard-max Q-values.

# 6 Related Work

Connections between softmax Q-values and optimal entropy-regularized policies have been previously noted. In some cases entropy regularization is expressed in the form of relative entropy [4, 6, 7, 31], and in other cases it is the standard entropy [47]. While these papers derive similar relationships to (7) and (8), they stop short of stating the single- and multi-step consistencies over all action choices we highlight. Moreover, the algorithms proposed in those works are essentially single-step Q-learning variants, which suffer from the limitation of using single-step backups. Another recent work [25] uses the softmax relationship in the limit of $\tau \to 0$ and proposes to augment an actor-critic algorithm with offline updates that minimize a set of single-step hard-max Bellman errors. Again, the methods we propose are differentiated by the multi-step path-wise consistencies which allow the resulting algorithms to utilize multi-step trajectories from off-policy samples in addition to on-policy samples.

The proposed PCL and Unified PCL algorithms bear some similarity to multi-step Q-learning [26], which rather than minimizing one-step hard-max Bellman error, optimizes a Q-value function approximator by unrolling the trajectory for some number of steps before using a hard-max backup. While this method has shown some empirical success [21], its theoretical justification is lacking, since rewards received after a non-optimal action no longer relate to the hard-max Q-values $Q^\circ$. In contrast, the algorithms we propose incorporate the log-probabilities of the actions on a multi-step rollout, which is crucial for the version of softmax consistency we consider.

Other notions of temporal consistency similar to softmax consistency have been discussed in the RL literature. Previous work has used a *Boltzmann weighted average* operator [20, 5]. In particular, this operator has been used by [5] to propose an iterative algorithm converging to the optimal maximum reward policy inspired by the work of [15, 39]. While they use the Boltzmann weighted average, they briefly mention that a softmax (log-sum-exp) operator would have similar theoretical properties. More recently [3] proposed a mellowmax operator, defined as log-*average*-exp. These log-average-exp operators share a similar non-expansion property, and the proofs of non-expansion are related.

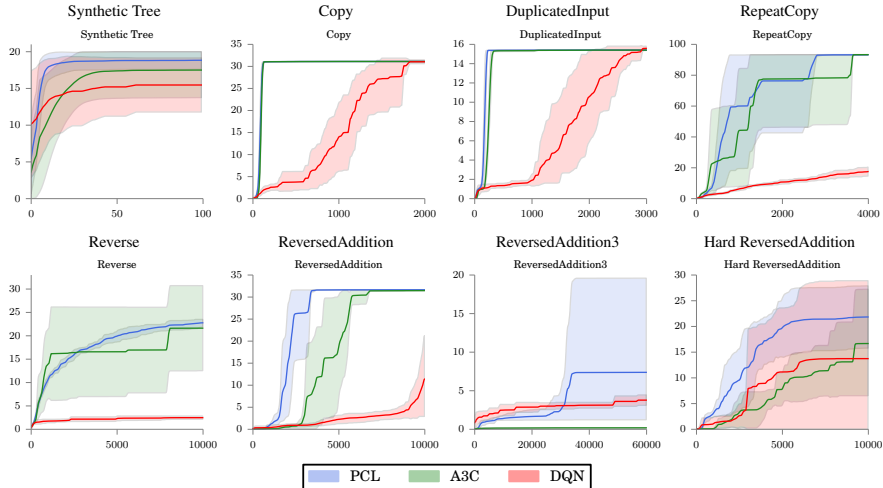

Figure 1: The results of PCL against A3C and DQN baselines. Each plot shows average reward across 5 random training runs (10 for Synthetic Tree) after choosing best hyperparameters. We also show a single standard deviation bar clipped at the min and max. The x-axis is number of training iterations. PCL exhibits comparable performance to A3C in some tasks, but clearly outperforms A3C on the more challenging tasks. Across all tasks, the performance of DQN is worse than PCL.

Additionally it is possible to show that when restricted to an infinite horizon setting, the fixed point of the mellowmax operator is a constant shift of the $Q^*$ investigated here. In all these cases, the suggested training algorithm optimizes a single-step consistency unlike PCL and Unified PCL, which optimizes a multi-step consistency. Moreover, these papers do not present a clear relationship between the action values at the fixed point and the entropy regularized expected reward objective, which was key to the formulation and algorithmic development in this paper.

Finally, there has been a considerable amount of work in reinforcement learning using off-policy data to design more sample efficient algorithms. Broadly speaking, these methods can be understood as trading off bias [36, 34, 19, 9] and variance [28, 23]. Previous work that has considered multi-step off-policy learning has typically used a correction (*e.g.,* via importance-sampling [29] or truncated importance sampling with bias correction [23], or eligibility traces [28]). By contrast, our method defines an unbiased consistency for an entire trajectory applicable to on- and off-policy data. An empirical comparison with all these methods remains however an interesting avenue for future work.

## 7 Experiments

We evaluate the proposed algorithms, namely PCL & Unified PCL, across several different tasks and compare them to an A3C implementation, based on [21], and an implementation of double Q-learning with prioritized experience replay, based on [30]. We find that PCL can consistently match or beat the performance of these baselines. We also provide a comparison between PCL and Unified PCL and find that the use of a single unified model for both values and policy can be competitive with PCL.

These new algorithms are easily amenable to incorporate expert trajectories. Thus, for the more difficult tasks we also experiment with seeding the replay buffer with 10 randomly sampled expert trajectories. During training we ensure that these trajectories are not removed from the replay buffer and always have a maximal priority.

The details of the tasks and the experimental setup are provided in the Appendix.

### 7.1 Results

We present the results of each of the variants PCL, A3C, and DQN in Figure 1. After finding the best hyperparameters (see the Supplementary Material), we plot the average reward over training iterations for five randomly seeded runs. For the Synthetic Tree environment, the same protocol is performed but with ten seeds instead.

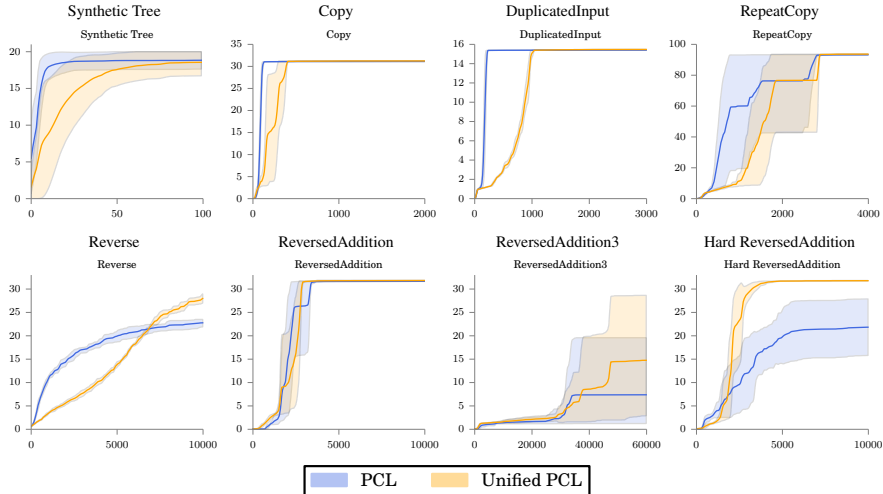

Figure 2: The results of PCL *vs.* Unified PCL. Overall we find that using a single model for both values and policy is not detrimental to training. Although in some of the simpler tasks PCL has an edge over Unified PCL, on the more difficult tasks, Unified PCL preforms better.

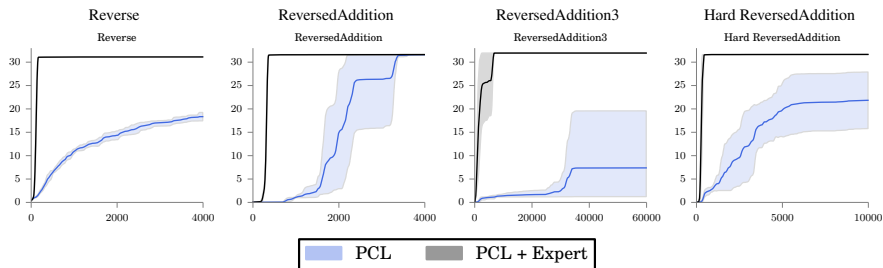

Figure 3: The results of PCL *vs.* PCL augmented with a small number of expert trajectories on the hardest algorithmic tasks. We find that incorporating expert trajectories greatly improves performance.

The gap between PCL and A3C is hard to discern in some of the more simple tasks such as Copy, Reverse, and RepeatCopy. However, a noticeable gap is observed in the Synthetic Tree and DuplicatedInput results and more significant gaps are clear in the harder tasks, including ReversedAddition, ReversedAddition3, and Hard ReversedAddition. Across all of the experiments, it is clear that the prioritized DQN performs worse than PCL. These results suggest that PCL is a competitive RL algorithm, which in some cases significantly outperforms strong baselines.

We compare PCL to Unified PCL in Figure 2. The same protocol is performed to find the best hyperparameters and plot the average reward over several training iterations. We find that using a single model for both values and policy in Unified PCL is slightly detrimental on the simpler tasks, but on the more difficult tasks Unified PCL is competitive or even better than PCL.

We present the results of PCL along with PCL augmented with expert trajectories in Figure 3. We observe that the incorporation of expert trajectories helps a considerable amount. Despite only using a small number of expert trajectories (*i.e.,* 10) as opposed to the mini-batch size of 400, the inclusion of expert trajectories in the training process significantly improves the agent's performance. We performed similar experiments with Unified PCL and observed a similar lift from using expert trajectories. Incorporating expert trajectories in PCL is relatively trivial compared to the specialized methods developed for other policy based algorithms [1, 12]. While we did not compare to other algorithms that take advantage of expert trajectories, this success shows the promise of using path-wise consistencies. Importantly, the ability of PCL to incorporate expert trajectories without requiring adjustment or correction is a desirable property in real-world applications such as robotics.

## 8 Conclusion

We study the characteristics of the optimal policy and state values for a maximum expected reward objective in the presence of *discounted entropy regularization*. The introduction of an entropy regularizer induces an interesting softmax consistency between the optimal policy and optimal state values, which may be expressed as either a single-step or multi-step consistency. This softmax consistency leads to the development of Path Consistency Learning (PCL), an RL algorithm that resembles actor-critic in that it maintains and jointly learns a model of the state values and a model of the policy, and is similar to Q-learning in that it minimizes a measure of temporal consistency error. We also propose the variant Unified PCL which maintains a single model for both the policy and the values, thus upending the actor-critic paradigm of separating the actor from the critic. Unlike standard policy based RL algorithms, PCL and Unified PCL apply to both on-policy and off-policy trajectory samples. Further, unlike value based RL algorithms, PCL and Unified PCL can take advantage of multi-step consistencies. Empirically, PCL and Unified PCL exhibit a significant improvement over baseline methods across several algorithmic benchmarks.

## 9 Acknowledgment

We thank Rafael Cosman, Brendan O'Donoghue, Volodymyr Mnih, George Tucker, Irwan Bello, and the Google Brain team for insightful comments and discussions.

## Footnotes

[1]Work done as a member of the Google Brain Residency program (`g.co/brainresidency`)

[2]An implementation of PCL can be found at `https://github.com/tensorflow/models/tree/master/research/pcl_rl`

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
