[Supplementary Material]

# Bridging the Gap Between Value and Policy Based Reinforcement Learning
## (Supplementary Material)

## A    Pseudocode

Pseudocode for PCL is presented in Algorithm 1.

## B    Experimental Details

We describe the tasks we experimented on as well as details of the experimental setup.

### B.1    Synthetic Tree

As an initial testbed, we developed a simple synthetic environment. The environment is defined by a binary decision tree of depth 20. For each training run, the reward on each edge is sampled uniformly from $[-1, 1]$ and subsequently normalized so that the maximal reward trajectory has total reward 20. We trained using a fully-parameterized model: for each node $s$ in the decision tree there are two parameters to determine the logits of $\pi_\theta(-|s)$ and one parameter to determine $V_\phi(s)$. In the Q-learning and Unified PCL implementations only two parameters per node $s$ are needed to determine the Q-values.

---

**Algorithm 1** Path Consistency Learning

> **Input:** Environment $ENV$, learning rates $\eta_\pi, \eta_v$, discount factor $\gamma$, rollout $d$, number of steps $N$, replay buffer capacity $B$, prioritized replay hyperparameter $\alpha$.
> **function** Gradients($s_{0:T}$)
>     *// We use $G(s_{t:t+d}, \pi_\theta)$ to denote a discounted sum of log-probabilities from $s_t$ to $s_{t+d}$.*
>     Compute $\Delta\theta = \sum_{t=0}^{T-d} C_{\theta,\phi}(s_{t:t+d})\nabla_\theta G(s_{t:t+d}, \pi_\theta)$.
>     Compute $\Delta\phi = \sum_{t=0}^{T-d} C_{\theta,\phi}(s_{t:t+d})\left(\nabla_\phi V_\phi(s_t) - \gamma^d \nabla_\phi V_\phi(s_{t+d})\right)$.
>     *Return $\Delta\theta, \Delta\phi$*
> **end function**
> Initialize $\theta, \phi$.
> Initialize empty replay buffer $RB(\alpha)$.
> **for** $i = 0$ **to** $N - 1$ **do**
>     Sample $s_{0:T} \sim \pi_\theta(s_{0:})$ on $ENV$.
>     $\Delta\theta, \Delta\phi = $ Gradients($s_{0:T}$).
>     Update $\theta \leftarrow \theta + \eta_\pi \Delta\theta$.
>     Update $\phi \leftarrow \phi + \eta_V \Delta\phi$.
>     Input $s_{0:T}$ into $RB$ with priority $R^1(s_{0:T})$.
>     If $|RB| > B$, remove episodes uniformly at random.
>     Sample $s_{0:T}$ from $RB$.
>     $\Delta\theta, \Delta\phi = $ Gradients($s_{0:T}$).
>     Update $\theta \leftarrow \theta + \eta_\pi \Delta\theta$.
>     Update $\phi \leftarrow \phi + \eta_v \Delta\phi$.
> **end for**

---

### B.2 Algorithmic Tasks

For more complex environments, we evaluated PCL, Unified PCL, and the two baselines on the algorithmic tasks from the OpenAI Gym library [4]. This library provides six tasks, in rough order of difficulty: Copy, DuplicatedInput, RepeatCopy, Reverse, ReversedAddition, and ReversedAddition3. In each of these tasks, an agent operates on a grid of characters or digits, observing one character or digit at a time. At each time step, the agent may move one step in any direction and optionally write a character or digit to output. A reward is received on each correct emission. The agent's goal for each task is:

- **Copy:** Copy a $1 \times n$ sequence of characters to output.
- **DuplicatedInput:** Deduplicate a $1 \times n$ sequence of characters.
- **RepeatCopy:** Copy a $1 \times n$ sequence of characters first in forward order, then reverse, and finally forward again.
- **Reverse:** Copy a $1 \times n$ sequence of characters in reverse order.
- **ReversedAddition:** Observe two ternary numbers in little-endian order via a $2 \times n$ grid and output their sum.
- **ReversedAddition3:** Observe three ternary numbers in little-endian order via a $3 \times n$ grid and output their sum.

These environments have an implicit curriculum associated with them. To observe the performance of our algorithm without curriculum, we also include a task "Hard ReversedAddition" which has the same goal as ReversedAddition but does not utilize curriculum.

For these environments, we parameterized the agent by a recurrent neural network with LSTM [5] cells of hidden dimension 128.

### B.3 Implementation Details

For our hyperparameter search, we found it simple to parameterize the critic learning rate in terms of the actor learning rate as $\eta_v = C\eta_\pi$, where $C$ is the *critic weight*.

For the Synthetic Tree environment we used a batch size of 10, rollout of $d = 3$, discount of $\gamma = 1.0$, and a replay buffer capacity of 10,000. We fixed the $\alpha$ parameter for PCL's replay buffer to 1 and used $\epsilon = 0.05$ for DQN. To find the optimal hyperparameters, we performed an extensive grid search over actor learning rate $\eta_\pi \in \{0.01, 0.05, 0.1\}$; critic weight $C \in \{0.1, 0.5, 1\}$; entropy regularizer $\tau \in \{0.005, 0.01, 0.025, 0.05, 0.1, 0.25, 0.5, 1.0\}$ for A3C, PCL, Unified PCL; and $\alpha \in \{0.1, 0.3, 0.5, 0.7, 0.9\}, \beta \in \{0.2, 0.4, 0.6, 0.8, 1.0\}$ for DQN replay buffer parameters. We used standard gradient descent for optimization.

For the algorithmic tasks we used a batch size of 400, rollout of $d = 10$, a replay buffer of capacity 100,000, ran using distributed training with 4 workers, and fixed the actor learning rate $\eta_\pi$ to 0.005, which we found to work well across all variants. To find the optimal hyperparameters, we performed an extensive grid search over discount $\gamma \in \{0.9, 1.0\}$, $\alpha \in \{0.1, 0.5\}$ for PCL's replay buffer; critic weight $C \in \{0.1, 1\}$; entropy regularizer $\tau \in \{0.005, 0.01, 0.025, 0.05, 0.1, 0.15\}$; $\alpha \in \{0.2, 0.4, 0.6, 0.8\}$, $\beta \in \{0.06, 0.2, 0.4, 0.5, 0.8\}$ for the prioritized DQN replay buffer; and also experimented with exploration rates $\epsilon \in \{0.05, 0.1\}$ and copy frequencies for the target DQN, $\{100, 200, 400, 600\}$. In these experiments, we used the Adam optimizer [6].

All experiments were implemented using Tensorflow [1].

## C Proofs

In this section, we provide a general theoretical foundation for this work, including proofs of the main path consistency results. We first establish the basic results for a simple one-shot decision making setting. These initial results will be useful in the proof of the general infinite horizon setting.

Although the main paper expresses the main claims under an assumption of deterministic dynamics, this assumption is not necessary: we restricted attention to the deterministic case in the main body merely for clarity and ease of explanation. Given that in this appendix we provide the general

foundations for this work, we consider the more general stochastic setting throughout the later sections.

In particular, for the general stochastic, infinite horizon setting, we introduce and discuss the entropy regularized expected return $O_{ENT}$ and define a "softmax" operator $\mathcal{B}^*$ (analogous to the Bellman operator for hard-max Q-values). We then show the existence of a unique fixed point $V^*$ of $\mathcal{B}^*$, by establishing that the softmax Bellman operator ($\mathcal{B}^*$) is a contraction under the infinity norm. We then relate $V^*$ to the optimal value of the entropy regularized expected reward objective $O_{ENT}$, which we term $V^\dagger$. We are able to show that $V^* = V^\dagger$, as expected. Subsequently, we present a policy determined by $V^*$ that satisfies $V^*(s) = O_{ENT}(s, \pi^*)$. Then given the characterization of $\pi^*$ in terms of $V^*$, we establish the consistency property stated in Theorem 1 of the main text. Finally, we show that a consistent solution is optimal by satisfying the KKT conditions of the constrained optimization problem (establishing Theorem 4 of the main text).

### C.1   Basic results for one-shot entropy regularized optimization

For $\tau > 0$ and any vector $\mathbf{q} \in \mathbb{R}^n$, $n < \infty$, define the scalar valued function $F_\tau$ (the "softmax") by

$$F_\tau(\mathbf{q}) = \tau \log \left( \sum_{a=1}^n e^{q_a/\tau} \right) \tag{1}$$

and define the vector valued function $\mathbf{f}_\tau$ (the "soft indmax") by

$$\mathbf{f}_\tau(\mathbf{q}) = \frac{e^{\mathbf{q}/\tau}}{\sum_{a=1}^n e^{q_a/\tau}} = e^{(\mathbf{q}-F_\tau(\mathbf{q}))/\tau}, \tag{2}$$

where the exponentiation is component-wise. It is easy to verify that $\mathbf{f}_\tau = \nabla F_\tau$. Note that $\mathbf{f}_\tau$ maps any real valued vector to a probability vector. We denote the probability simplex by $\Delta = \{\pi : \pi \geq 0, \mathbf{1} \cdot \pi = 1\}$, and denote the entropy function by $H(\pi) = -\pi \cdot \log \pi$.

**Lemma 1.**

$$F_\tau(\mathbf{q}) = \max_{\pi \in \Delta} \left\{ \pi \cdot \mathbf{q} + \tau H(\pi) \right\} \tag{3}$$

$$= \mathbf{f}_\tau(\mathbf{q}) \cdot \mathbf{q} + \tau H(\mathbf{f}_\tau(\mathbf{q})) \tag{4}$$

*Proof.* First consider the constrained optimization problem on the right hand side of (3). The Lagrangian is given by $L = \pi \cdot (\mathbf{q} - \tau \log \pi) + \lambda(1 - \mathbf{1} \cdot \pi)$, hence $\nabla L = \mathbf{q} - \tau \log \pi - \tau - \lambda$. The KKT conditions for this optimization problems are the following system of $n + 1$ equations

$$\mathbf{1} \cdot \pi = 1 \tag{5}$$
$$\tau \log \pi = \mathbf{q} - v \tag{6}$$

for the $n + 1$ unknowns, $\pi$ and $v$, where $v = \lambda + \tau$. Note that for any $v$, satisfying (6) requires the unique assignment $\pi = \exp((\mathbf{q} - v)/\tau)$, which also ensures $\pi > 0$. To subsequently satisfy (5), the equation $1 = \sum_a \exp((q_a - v)/\tau) = e^{-v/\tau} \sum_a \exp(q_a/\tau)$ must be solved for $v$; since the right hand side is strictly decreasing in $v$, the solution is also unique and in this case given by $v = F_\tau(\mathbf{q})$. Therefore $\pi = \mathbf{f}_\tau(\mathbf{q})$ and $v = F_\tau(\mathbf{q})$ provide the unique solution to the KKT conditions (5)-(6). Since the objective is strictly concave, $\pi$ must be the unique global maximizer, establishing (4). It is then easy to show $F_\tau(\mathbf{q}) = \mathbf{f}_\tau(\mathbf{q}) \cdot \mathbf{q} + \tau H(\mathbf{f}_\tau(\mathbf{q}))$ by algebraic manipulation, which establishes (3). $\square$

**Corollary 2** (Optimality Implies Consistency). *If* $v^* = \max_{\pi \in \Delta} \left\{ \pi \cdot \mathbf{q} + \tau H(\pi) \right\}$ *then*

$$v^* = q_a - \tau \log \pi_a^* \text{ for all } a, \tag{7}$$

*where* $\pi^* = \mathbf{f}_\tau(\mathbf{q})$.

*Proof.* From Lemma 1 we know $v^* = F_\tau(\mathbf{q}) = \pi^* \cdot (\mathbf{q} - \tau \log \pi^*)$ where $\pi^* = \mathbf{f}_\tau(\mathbf{q})$. From the definition of $\mathbf{f}_\tau$ it also follows that $\log \pi_a^* = (q_a - F_\tau(\mathbf{q}))/\tau$ for all $a$, hence $v^* = F_\tau(\mathbf{q}) = q_a - \tau \log \pi_a^*$ for all $a$. $\square$

**Corollary 3** (Consistency Implies Optimality)**.** *If $v \in \mathbb{R}$ and $\boldsymbol{\pi} \in \Delta$ jointly satisfy*

$$v = q_a - \tau \log \pi_a \ \text{ for all } a, \tag{8}$$

*then $v = F_\tau(\mathbf{q})$ and $\boldsymbol{\pi} = \mathbf{f}_\tau(\mathbf{q})$; that is, $\boldsymbol{\pi}$ must be an optimizer for* (3) *and $v$ is its corresponding optimal value.*

*Proof.* Any $v$ and $\boldsymbol{\pi} \in \Delta$ that jointly satisfy (8) must also satisfy the KKT conditions (5)-(6); hence $\boldsymbol{\pi}$ must be the unique maximizer for (3) and $v$ its corresponding objective value. □

Although these results are elementary, they reveal a strong connection between optimal state values ($v$), optimal action values ($\mathbf{q}$) and optimal policies ($\boldsymbol{\pi}$) under the softmax operators. In particular, Lemma 1 states that, if $\mathbf{q}$ is an optimal action value at some current state, the optimal state value must be $v = F_\tau(\mathbf{q})$, which is simply the entropy regularized value of the optimal policy, $\boldsymbol{\pi} = \mathbf{f}_\tau(\mathbf{q})$, at the current state.

Corollaries 2 and 3 then make the stronger observation that this mutual consistency between the optimal state value, optimal action values and optimal policy probabilities must hold for *every* action, not just in expectation over actions sampled from $\boldsymbol{\pi}$; and furthermore that achieving mutual consistency in this form is *equivalent* to achieving optimality.

Below we will also need to make use of the following properties of $F_\tau$.

**Lemma 4.** *For any vector $\mathbf{q}$,*

$$F_\tau(\mathbf{q}) = \sup_{\mathbf{p} \in \Delta}\left\{\mathbf{p} \cdot \mathbf{q} - \tau \mathbf{p} \cdot \log \mathbf{p}\right\}. \tag{9}$$

*Proof.* Let $F_\tau^*$ denote the conjugate of $F_\tau$, which is given by

$$F_\tau^*(\mathbf{p}) = \sup_{\mathbf{q}}\left\{\mathbf{q} \cdot \mathbf{p} - F_\tau(\mathbf{q})\right\} = \tau \mathbf{p} \cdot \log \mathbf{p} \tag{10}$$

for $\mathbf{p} \in \operatorname{dom}(F_\tau^*) = \Delta$. Since $F_\tau$ is closed and convex, we also have that $F_\tau = F_\tau^{**}$ [3, Section 4.2]; hence

$$F_\tau(\mathbf{q}) = \sup_{\mathbf{p} \in \Delta}\left\{\mathbf{q} \cdot \mathbf{p} - F_\tau^*(\mathbf{p})\right\}. \tag{11}$$

□

**Lemma 5.** *For any two vectors $\mathbf{q}^{(1)}$ and $\mathbf{q}^{(2)}$,*

$$F_\tau(\mathbf{q}^{(1)}) - F_\tau(\mathbf{q}^{(2)}) \le \max_a\left\{q_a^{(1)} - q_a^{(2)}\right\}. \tag{12}$$

*Proof.* Observe that by Lemma 4

$$F_\tau(\mathbf{q}^{(1)}) - F_\tau(\mathbf{q}^{(2)}) = \sup_{\mathbf{p}^{(1)} \in \Delta}\left\{\mathbf{q}^{(1)} \cdot \mathbf{p}^{(1)} - F_\tau^*(\mathbf{p}^{(1)})\right\} - \sup_{\mathbf{p}^{(2)} \in \Delta}\left\{\mathbf{q}^{(2)} \cdot \mathbf{p}^{(2)} - F_\tau^*(\mathbf{p}^{(2)})\right\} \tag{13}$$

$$= \sup_{\mathbf{p}^{(1)} \in \Delta}\left\{\inf_{\mathbf{p}^{(2)} \in \Delta}\left\{\mathbf{q}^{(1)} \cdot \mathbf{p}^{(1)} - \mathbf{q}^{(2)} \cdot \mathbf{p}^{(2)} - (F_\tau^*(\mathbf{p}^{(1)}) - F_\tau^*(\mathbf{p}^{(2)}))\right\}\right\} \tag{14}$$

$$\le \sup_{\mathbf{p}^{(1)} \in \Delta}\left\{\mathbf{p}^{(1)} \cdot (\mathbf{q}^{(1)} - \mathbf{q}^{(2)})\right\} \quad \text{by choosing } \mathbf{p}^{(2)} = \mathbf{p}^{(1)} \tag{15}$$

$$\le \max_a\left\{q_a^{(1)} - q_a^{(2)}\right\}. \tag{16}$$

□

**Corollary 6.** *$F_\tau$ is an $\infty$-norm contraction; that is, for any two vectors $\mathbf{q}^{(1)}$ and $\mathbf{q}^{(2)}$,*

$$\left|F_\tau(\mathbf{q}^{(1)}) - F_\tau(\mathbf{q}^{(2)})\right| \le \|\mathbf{q}^{(1)} - \mathbf{q}^{(2)}\|_\infty \tag{17}$$

*Proof.* Immediate from Lemma 5. □

## C.2 Background results for *on-policy* entropy regularized updates

Although the results in the main body of the paper are expressed in terms of deterministic problems, we will prove that all the desired properties hold for the more general **stochastic** case, where there is a stochastic transition $s, a \mapsto s'$ determined by the environment. Given the characterization for this general case, the application to the deterministic case is immediate. We continue to assume that the action space is finite, and that the state space is discrete.

For any policy $\pi$, define the entropy regularized expected return by

$$\tilde{V}^\pi(s_\ell) = O_{\text{ENT}}(s_\ell, \pi) = \mathbb{E}_{a_\ell s_{\ell+1} \ldots | s_\ell} \left[ \sum_{i=0}^{\infty} \gamma^i \big( r(s_{\ell+i}, a_{\ell+i}) - \tau \log \pi(a_{\ell+i}|s_{\ell+i}) \big) \right], \qquad (18)$$

where the expectation is taken with respect to the policy $\pi$ and with respect to the stochastic state transitions determined by the environment. We will find it convenient to also work with the on-policy Bellman operator defined by

$$(\mathcal{B}^\pi V)(s) = \mathbb{E}_{a, s'|s} \left[ r(s, a) - \tau \log \pi(a|s) + \gamma V(s') \right] \qquad (19)$$

$$= \mathbb{E}_{a|s} \left[ r(s, a) - \tau \log \pi(a|s) + \gamma \mathbb{E}_{s'|s,a} \big[ V(s') \big] \right] \qquad (20)$$

$$= \pi(:|s) \cdot (Q(s,:) - \tau \log \pi(:|s)), \quad \text{where} \qquad (21)$$

$$Q(s, a) = r(s, a) + \gamma \mathbb{E}_{s'|s,a}[V(s')] \qquad (22)$$

for each state $s$ and action $a$. Note that in (21) we are using $Q(s,:)$ to denote a vector values over choices of $a$ for a given $s$, and $\pi(:|s)$ to denote the vector of conditional action probabilities specified by $\pi$ at state $s$.

**Lemma 7.** *For any policy $\pi$ and state $s$, $\tilde{V}^\pi(s)$ satisfies the recurrence*

$$\tilde{V}^\pi(s) = \mathbb{E}_{a|s} \left[ r(s, a) + \gamma \mathbb{E}_{s'|s,a}[\tilde{V}^\pi(s')] - \tau \log \pi(a|s) \right] \qquad (23)$$

$$= \pi(:|s) \cdot \left( \tilde{Q}^\pi(s,:) - \tau \log \pi(:|s) \right) \text{ where } \tilde{Q}^\pi(s, a) = r(s, a) + \gamma \mathbb{E}_{s'|s,a}[\tilde{V}^\pi(s')] \quad (24)$$

$$= (\mathcal{B}^\pi \tilde{V}^\pi)(s). \qquad (25)$$

*Moreover, $\mathcal{B}^\pi$ is a contraction mapping.*

*Proof.* Consider an arbitrary state $s_\ell$. By the definition of $\tilde{V}^\pi(s_\ell)$ in (18) we have

$$\tilde{V}^\pi(s_\ell) = \mathbb{E}_{a_\ell s_{\ell+1} \ldots | s_\ell} \left[ \sum_{i=0}^{\infty} \gamma^i \big( r(s_{\ell+i}, a_{\ell+i}) - \tau \log \pi(a_{\ell+i}|s_{\ell+i}) \big) \right] \qquad (26)$$

$$= \mathbb{E}_{a_\ell s_{\ell+1} \ldots | s_\ell} \Bigg[ r(s_\ell, a_\ell) - \tau \log \pi(a_\ell|s_\ell) \qquad (27)$$

$$+ \gamma \sum_{j=0}^{\infty} \gamma^j \big( r(s_{\ell+1+j}, a_{\ell+1+j}) - \tau \log \pi(a_{\ell+1+j}|s_{\ell+1+j}) \big) \Bigg]$$

$$= \mathbb{E}_{a_\ell|s_\ell} \Bigg[ r(s_\ell, a_\ell) - \tau \log \pi(a_\ell|s_\ell) \qquad (28)$$

$$+ \gamma \mathbb{E}_{s_{\ell+1} a_{\ell+1} \ldots | s_\ell, a_\ell} \left[ \sum_{j=0}^{\infty} \gamma^j \big( r(s_{\ell+1+j}, a_{\ell+1+j}) - \tau \log \pi(a_{\ell+1+j}|s_{\ell+1+j}) \big) \right] \Bigg]$$

$$= \mathbb{E}_{a_\ell|s_\ell} \left[ r(s_\ell, a_\ell) - \tau \log \pi(a_\ell|s_\ell) + \gamma \mathbb{E}_{s_{\ell+1}|s_\ell, a_\ell} [\tilde{V}^\pi(s_{\ell+1})] \right] \qquad (29)$$

$$= \pi(:|s_\ell) \cdot \left( \tilde{Q}^\pi(s_\ell, :) - \tau \log \pi(:|s_\ell) \right) \qquad (30)$$

$$= (\mathcal{B}^\pi \tilde{V}^\pi)(s_\ell). \qquad (31)$$

The fact that $\mathcal{B}^\pi$ is a contraction mapping follows directly from standard arguments about the on-policy Bellman operator [7]. $\qquad \square$

Note that this lemma shows $\tilde{V}^{\pi}$ is a *fixed point* of the corresponding on-policy Bellman operator $\mathcal{B}^{\pi}$. Next, we characterize how quickly convergence to a fixed point is achieved by repeated application of ther $\mathcal{B}^{\pi}$ operator.

**Lemma 8.** *For any $\pi$ and any $V$, for all states $s_{\ell}$, and for all $k \geq 0$ it holds that:*
$$\left((\mathcal{B}^{\pi})^k V\right)(s_{\ell}) - \tilde{V}^{\pi}(s_{\ell}) = \gamma^k \mathbb{E}_{a_{\ell} s_{\ell+1} \ldots s_{\ell+k} | s_{\ell}} \left[ V(s_{\ell+k}) - \tilde{V}^{\pi}(s_{\ell+k}) \right].$$

*Proof.* Consider an arbitrary state $s_{\ell}$. We use an induction on $k$. For the base case, consider $k = 0$ and observe that the claim follows trivially, since $\left((\mathcal{B}^{\pi})^0 V\right)(s_{\ell}) - \left((\mathcal{B}^{\pi})^0 \tilde{V}^{\pi}\right)(s_{\ell}) = V(s_{\ell}) - \tilde{V}^{\pi}(s_{\ell})$. For the induction hypothesis, assume the result holds for $k$. Then consider:

$$
\begin{aligned}
&\left((\mathcal{B}^{\pi})^{k+1} V\right)(s_{\ell}) - \left(\tilde{V}^{\pi}\right)(s_{\ell}) \\
&= \left((\mathcal{B}^{\pi})^{k+1} V\right)(s_{\ell}) - \left((\mathcal{B}^{\pi})^{k+1} \tilde{V}^{\pi}\right)(s_{\ell}) \quad \text{(by Lemma 7)} &(32)\\
&= \left(\mathcal{B}^{\pi}(\mathcal{B}^{\pi})^k V\right)(s_{\ell}) - \left(\mathcal{B}^{\pi}(\mathcal{B}^{\pi})^k \tilde{V}^{\pi}\right)(s_{\ell}) &(33)\\
&= \mathbb{E}_{a_{\ell} s_{\ell+1} | s_{\ell}} \left[ r(s_{\ell}, a_{\ell}) - \tau \log \pi(a_{\ell} | s_{\ell}) + \gamma (\mathcal{B}^{\pi})^k V(s_{\ell+1}) \right] \\
&\quad - \mathbb{E}_{a_{\ell} s_{\ell+1} | s_{\ell}} \left[ r(s_{\ell}, a_{\ell}) - \tau \log \pi(a_{\ell} | s_{\ell}) + \gamma (\mathcal{B}^{\pi})^k \tilde{V}^{\pi}(s_{\ell+1}) \right] &(34)\\
&= \gamma \mathbb{E}_{a_{\ell} s_{\ell+1} | s_{\ell}} \left[ (\mathcal{B}^{\pi})^k V(s_{\ell+1}) - (\mathcal{B}^{\pi})^k \tilde{V}^{\pi}(s_{\ell+1}) \right] &(35)\\
&= \gamma \mathbb{E}_{a_{\ell} s_{\ell+1} | s_{\ell}} \left[ (\mathcal{B}^{\pi})^k V(s_{\ell+1}) - \tilde{V}^{\pi}(s_{\ell+1}) \right] \quad \text{(by Lemma 7)} &(36)\\
&= \gamma \mathbb{E}_{a_{\ell} s_{\ell+1} | s_{\ell}} \left[ \gamma^k \mathbb{E}_{a_{\ell+1} s_{\ell+2} \ldots s_{\ell+k+1} | s_{\ell+1}} \left[ V(s_{\ell+k+1}) - \tilde{V}^{\pi}(s_{\ell+k+1}) \right] \right] \quad \text{(by IH)} &(37)\\
&= \gamma^{k+1} \mathbb{E}_{a_{\ell} s_{\ell+1} \ldots s_{\ell+k+1} | s_{\ell}} \left[ V(s_{\ell+k+1}) - \tilde{V}^{\pi}(s_{\ell+k+1}) \right], &(38)
\end{aligned}
$$

establishing the claim. $\square$

**Lemma 9.** *For any $\pi$ and any $V$, we have $\left\| (\mathcal{B}^{\pi})^k V - \tilde{V}^{\pi} \right\|_{\infty} \leq \gamma^k \left\| V - \tilde{V}^{\pi} \right\|_{\infty}$.*

*Proof.* Let $p^{(k)}(s_{\ell+k} | s_{\ell})$ denote the conditional distribution over the $k$th state, $s_{\ell+k}$, visited in a random walk starting from $s_{\ell}$, which is induced by the environment and the policy $\pi$. Consider

$$
\begin{aligned}
\left\| (\mathcal{B}^{\pi})^k V - \tilde{V}^{\pi} \right\|_{\infty} &= \gamma^k \max_{s_{\ell}} \left| \mathbb{E}_{a_{\ell} s_{\ell+1} \ldots s_{\ell+k} | s_{\ell}} \left[ V(s_{\ell+k}) - \tilde{V}^{\pi}(s_{\ell+k}) \right] \right| \quad \text{(by Lemma 8)} &(39)\\
&= \gamma^k \max_{s_{\ell}} \left| \sum_{s_{\ell+k}} p^{(k)}(s_{\ell+k} | s_{\ell}) \left( V(s_{\ell+k}) - \tilde{V}^{\pi}(s_{\ell+k}) \right) \right| &(40)\\
&= \gamma^k \max_{s_{\ell}} \left| p^{(k)}(: | s_{\ell}) \cdot \left( V - \tilde{V}^{\pi} \right) \right| &(41)\\
&\leq \gamma^k \max_{s_{\ell}} \| p^{(k)}(: | s_{\ell}) \|_1 \| V - \tilde{V}^{\pi} \|_{\infty} \quad \text{(by Hölder's inequality)} &(42)\\
&= \gamma^k \| V - \tilde{V}^{\pi} \|_{\infty}. &(43)
\end{aligned}
$$

$\square$

**Corollary 10.** *For any bounded $V$ and any $\epsilon > 0$ there exists a $k_0$ such that $(\mathcal{B}^{\pi})^k V \geq \tilde{V}^{\pi} - \epsilon$ for all $k \geq k_0$.*

*Proof.* By Lemma 9 we have $(\mathcal{B}^{\pi})^k V \geq \tilde{V}^{\pi} - \gamma^k \left\| V - \tilde{V}^{\pi} \right\|_{\infty}$ for all $k \geq 0$. Therefore, for any $\epsilon > 0$ there exists a $k_0$ such that $\gamma^k \left\| V - \tilde{V}^{\pi} \right\|_{\infty} < \epsilon$ for all $k \geq k_0$, since $V$ is assumed bounded. $\square$

Thus, any value function will converge to $\tilde{V}^{\pi}$ via repeated application of on-policy backups $\mathcal{B}^{\pi}$. Below we will also need to make use of the following monotonicity property of the on-policy Bellman operator.

**Lemma 11.** *For any $\pi$, if $V^{(1)} \geq V^{(2)}$ then $\mathcal{B}^{\pi} V^{(1)} \geq \mathcal{B}^{\pi} V^{(2)}$.*

*Proof.* Assume $V^{(1)} \geq V^{(2)}$ and note that for any state $s_\ell$

$$(\mathcal{B}^\pi V^{(2)})(s_\ell) - (\mathcal{B}^\pi V^{(1)})(s_\ell) = \gamma \mathbb{E}_{a_\ell s_{\ell+1}|s_\ell} \left[ V^{(2)}(s_{\ell+1}) - V^{(1)}(s_{\ell+1}) \right] \tag{44}$$

$$\leq 0 \quad \text{since it was assumed that } V^{(2)} \leq V^{(1)}. \tag{45}$$

$\square$

## C.3   Proof of main optimality claims for *off-policy* softmax updates

Define the optimal value function by

$$V^\dagger(s) = \max_\pi O_{\text{ENT}}(s, \pi) = \max_\pi \tilde{V}^\pi(s) \text{ for all } s. \tag{46}$$

For $\tau > 0$, define the softmax Bellman operator $\mathcal{B}^*$ by

$$(\mathcal{B}^*V)(s) = \tau \log \sum_a \exp \left( \left( r(s, a) + \gamma \mathbb{E}_{s'|s,a}[V(s')] \right)/\tau \right) \tag{47}$$

$$= F_\tau(Q(s,:)) \quad \text{where} \quad Q(s, a) = r(s, a) + \gamma \mathbb{E}_{s'|s,a}[V(s')] \quad \text{for all } a. \tag{48}$$

**Lemma 12.** *For $\gamma < 1$, the fixed point of the softmax Bellman operator, $V^* = \mathcal{B}^*V^*$, exists and is unique.*

*Proof.* First observe that the softmax Bellman operator is a contraction in the infinity norm. That is, consider two value functions, $V^{(1)}$ and $V^{(2)}$, and let $p(s'|s, a)$ denote the state transition probability function determined by the environment. We then have

$$\left\| \mathcal{B}^*V^{(1)} - \mathcal{B}^*V^{(2)} \right\|_\infty = \max_s \left| (\mathcal{B}^*V^{(1)})(s) - (\mathcal{B}^*V^{(2)})(s) \right| \tag{49}$$

$$= \max_s \left| F_\tau(Q^{(1)}(s,:)) - F_\tau(Q^{(2)}(s,:)) \right| \tag{50}$$

$$\leq \max_s \max_a \left| Q^{(1)}(s, a) - Q^{(2)}(s, a) \right| \quad \text{(by Corollary 6)} \tag{51}$$

$$= \gamma \max_s \max_a \left| \mathbb{E}_{s'|s,a} \left[ V^{(1)}(s') - V^{(2)}(s') \right] \right| \tag{52}$$

$$= \gamma \max_s \max_a \left| p(:|s, a) \cdot \left( V^{(1)} - V^{(2)} \right) \right| \tag{53}$$

$$\leq \gamma \max_s \max_a \| p(:|s, a) \|_1 \| V^{(1)} - V^{(2)} \|_\infty \quad \text{(Hölder's inequality)} \tag{54}$$

$$= \gamma \| V^{(1)} - V^{(2)} \|_\infty < \| V^{(1)} - V^{(2)} \|_\infty \text{ if } \gamma < 1. \tag{55}$$

The existence and uniqueness of $V^*$ then follows from the contraction map fixed-point theorem [2]. $\square$

**Lemma 13.** *For any $\pi$, if $V \geq \mathcal{B}^*V$ then $V \geq (\mathcal{B}^\pi)^k V$ for all $k$.*

*Proof.* Observe for any $s$ that the assumption implies

$$V(s) \geq (\mathcal{B}^*V)(s) \tag{56}$$

$$= \max_{\tilde{\pi}(:|s) \in \Delta} \sum_a \tilde{\pi}(a|s) \left( r(s, a) + \gamma \mathbb{E}_{s'|s,a}[V(s')] - \tau \log \tilde{\pi}(a|s) \right) \tag{57}$$

$$\geq \sum_a \pi(a|s) \left( r(s, a) + \gamma \mathbb{E}_{s'|s,a}[V(s')] - \tau \log \pi(a|s) \right) \tag{58}$$

$$= (\mathcal{B}^\pi V)(s). \tag{59}$$

The result then follows by the monotonicity of $\mathcal{B}^\pi$ (Lemma 11). $\square$

**Corollary 14.** *For any $\pi$, if $V$ is bounded and $V \geq \mathcal{B}^*V$, then $V \geq \tilde{V}^\pi$.*

*Proof.* Consider an arbitrary policy $\pi$. If $V \geq \mathcal{B}^*V$, then by Corollary 14 we have $V \geq (\mathcal{B}^\pi)^k V$ for all $k$. Then by Corollary 10, for any $\epsilon > 0$ there exists a $k_0$ such that $V \geq (\mathcal{B}^\pi)^k V \geq \tilde{V}^\pi - \epsilon$ for all $k \geq k_0$ since $V$ is bounded; hence $V \geq \tilde{V}^\pi - \epsilon$ for all $\epsilon > 0$. We conclude that $V \geq \tilde{V}^\pi$. $\square$

Next, given the existence of $V^*$, we define a specific policy $\pi^*$ as follows

$$\pi^*(:|s) = \mathbf{f}_\tau\big(Q^*(s,:)\big), \quad \text{where} \tag{60}$$

$$Q^*(s,a) = r(s,a) + \gamma \mathbb{E}_{s'|s,a}[V^*(s')]. \tag{61}$$

Note that we are simply defining $\pi^*$ at this stage and have not as yet proved it has any particular properties; but we will see shortly that it is, in fact, an optimal policy.

**Lemma 15.** $V^* = \tilde{V}^{\pi^*}$; that is, for $\pi^*$ defined in (60), $V^*$ gives its entropy regularized expected return from any state.

*Proof.* We establish the claim by showing $\mathcal{B}^* \tilde{V}^{\pi^*} = \tilde{V}^{\pi^*}$. In particular, for an arbitrary state $s$ consider

$$(\mathcal{B}^* \tilde{V}^{\pi^*})(s) = F_\tau\big(\tilde{Q}^{\pi^*}(s,:)\big) \qquad \text{by (48)} \tag{62}$$

$$= \pi^*(:|s) \cdot \big(\tilde{Q}^{\pi^*}(s,:) - \tau \log \pi^*(:|s)\big) \qquad \text{by Lemma 1} \tag{63}$$

$$= \tilde{V}^{\pi^*}(s) \qquad \text{by Lemma 7.} \tag{64}$$

$\square$

**Theorem 16.** *The fixed point of the softmax Bellman operator is the optimal value function:* $V^* = V^\dagger$.

*Proof.* Since $V^* \geq \mathcal{B}^* V^*$ (in fact, $V^* = \mathcal{B}^* V^*$) we have $V^* \geq \tilde{V}^\pi$ for all $\pi$ by Corollary 14, hence $V^* \geq V^\dagger$. Next observe that by Lemma 15 we have $V^\dagger \geq \tilde{V}^{\pi^*} = V^*$. Finally, by Lemma 12, we know that the fixed point $V^* = \mathcal{B}^* V^*$ is unique, hence $V^\dagger = V^*$. $\square$

**Corollary 17** (Optimality Implies Consistency). *The optimal state value function* $V^*$ *and optimal policy* $\pi^*$ *satisfy* $V^*(s) = r(s,a) + \gamma \mathbb{E}_{s'|s,a}[V^*(s')] - \tau \log \pi^*(a|s)$ *for every state* $s$ *and action* $a$.

*Proof.* First note that

$$Q^*(s,a) = r(s,a) + \gamma \mathbb{E}_{s'|s,a}[V^*(s')] \qquad \text{by (61)} \tag{65}$$

$$= r(s,a) + \gamma \mathbb{E}_{s'|s,a}[V^{\pi^*}(s')] \qquad \text{by Lemma 15} \tag{66}$$

$$= Q^{\pi^*}(s,a) \qquad \text{by (22).} \tag{67}$$

Then observe that for any state $s$,

$$V^*(s) = F_\tau\big(Q^*(s,:)\big) \qquad \text{by (48)} \tag{68}$$

$$= F_\tau\big(Q^{\pi^*}(s,:)\big) \qquad \text{from above} \tag{69}$$

$$= \pi^*(:|s) \cdot \big(Q^{\pi^*}(s,:) - \tau \log \pi^*(:|s)\big) \qquad \text{by Lemma 1} \tag{70}$$

$$= Q^{\pi^*}(s,a) - \tau \log \pi^*(a|s) \text{ for all } a \qquad \text{by Corollary 2} \tag{71}$$

$$= Q^*(s,a) - \tau \log \pi^*(a|s) \text{ for all } a \qquad \text{from above.} \tag{72}$$

$\square$

**Corollary 18** (Consistency Implies Optimality). *If* $V$ *and* $\pi$ *satisfy, for all* $s$ *and* $a$: $V(s) = r(s,a) + \gamma \mathbb{E}_{s'|s,a}[V(s')] - \tau \log \pi(a|s)$; *then* $V = V^*$ *and* $\pi = \pi^*$.

*Proof.* We will show that satisfying the constraint for every $s$ and $a$ implies $\mathcal{B}^* V = V$; it will then immediately follow that $V = V^*$ and $\pi = \pi^*$ by Lemma 12. Let $Q(s,a) = r(s,a) + \gamma \mathbb{E}_{s'|s,a}[V(s')]$. Consider an arbitrary state $s$, and observe that

$$(\mathcal{B}^* V)(s) = F_\tau\big(Q(s,:)\big) \quad \text{(by (48))} \tag{73}$$

$$= \max_{\boldsymbol{\pi} \in \Delta}\Big\{\boldsymbol{\pi} \cdot \big(Q(s,:) - \tau \log \boldsymbol{\pi}\big)\Big\} \quad \text{(by Lemma 1)} \tag{74}$$

$$= Q(s,a) - \tau \log \pi(a|s) \text{ for all } a \quad \text{(by Corollary 3)} \tag{75}$$

$$= r(s,a) + \gamma \mathbb{E}_{s'|s,a}[V(s')] - \tau \log \pi(a|s) \text{ for all } a \quad \text{(by definition of } Q \text{ above)} \tag{76}$$

$$= V(s) \quad \text{(by the consistency assumption on } V \text{ and } \pi\text{).} \tag{77}$$

$\square$

## C.4  Proof of Theorem 1 from Main Text

**Note**: Theorem 1 from the main body was stated under an assumption of deterministic dynamics. We used this assumption in the main body merely to keep presentation simple and understandable. The development given in this appendix considers the more general case of a stochastic environment. We give the proof here for the more general setting; the result stated in Theorem 1 follows as a special case.

*Proof.* Assuming a stochastic environment, as developed in this appendix, we will establish that the optimal policy and state value function, $\pi^*$ and $V^*$ respectively, satisfy

$$V^*(s) = -\tau \log \pi^*(a|s) + r(s,a) + \gamma \mathbb{E}_{s'|s,a}[V^*(s')] \tag{78}$$

for all $s$ and $a$. Theorem 1 will then follow as a special case.

Consider the policy $\pi^*$ defined in (60). From Corollary 15 we know that $\tilde{V}^{\pi^*} = V^*$ and from Theorem 16 we know $V^* = V^\dagger$, hence $\tilde{V}^{\pi^*} = V^\dagger$; that is, $\pi^*$ is the optimizer of $O_{\text{ENT}}(s,\pi)$ for any state $s$ (including $s_0$). Therefore, this must be the same $\pi^*$ as considered in the premise. The assertion (78) then follows directly from Corollary 17. $\qquad\square$

## C.5  Proof of Corollary 2 from Main Text

**Note**: We consider the more general case of a stochastic environment as developed in this appendix. First note that the consistency property for the stochastic case (78) can be rewritten as

$$\mathbb{E}_{s'|s,a}\big[-V^*(s) + \gamma V^*(s') + r(s,a) - \tau \log \pi^*(a|s)\big] = 0 \tag{79}$$

for all $s$ and $a$. For a stochastic environment, the generalized version of (13) in Corollary 2 can then be expressed as

$$\mathbb{E}_{s_2\dots s_t|s_1,a_1\dots a_{t-1}}\left[-V^*(s_1) + \gamma^{t-1}V^*(s_t) + \sum_{i=1}^{t-1}\gamma^{i-1}\big(r(s_i,a_i) - \tau \log \pi^*(a_i|s_i)\big)\right] = 0 \tag{80}$$

for all states $s_1$ and action sequences $a_1\dots a_{t-1}$. We now show that (79) implies (80).

*Proof.* Observe that by (79) we have

$$0 = \mathbb{E}_{s_2\dots s_t|s_1,a_1\dots a_{t-1}}\left[\sum_{i=1}^{t-1}\gamma^{i-1}\big(-V^*(s_i) + \gamma V^*(s_{i+1}) + r(s_i,a_i) - \tau \log \pi^*(a_i|s_i)\big)\right] \tag{81}$$

$$= \mathbb{E}_{s_2\dots s_t|s_1,a_1\dots a_{t-1}}\left[\sum_{i=1}^{t-1}\gamma^{i-1}\big(-V^*(s_i) + \gamma V^*(s_{i+1})\big)\right.$$
$$\left. + \sum_{i=1}^{t-1}\gamma^{i-1}\big(r(s_i,a_i) - \tau \log \pi^*(a_i|s_i)\big)\right] \tag{82}$$

$$= \mathbb{E}_{s_2\dots s_t|s_1,a_1\dots a_{t-1}}\left[-V^*(s_1) + \gamma^{t-1}V^*(s_t) + \sum_{i=1}^{t-1}\gamma^{i-1}\big(r(s_i,a_i) - \tau \log \pi^*(a_i|s_i)\big)\right] \tag{83}$$

by a telescopic sum on the first term, which yields the result. $\qquad\square$

## C.6  Proof of Theorem 3 from Main Text

**Note**: Again, we consider the more general case of a stochastic environment. The consistency property in this setting is given by (78) above.

*Proof.* Consider a policy $\pi_\theta$ and value function $V_\phi$ that satisfy the general consistency property for a stochastic environment: $V_\phi(s) = -\tau \log \pi_\theta(a|s) + r(s,a) + \gamma \mathbb{E}_{s'|s,a}[V_\phi(s')]$ for all $s$ and $a$. Then by Corollary 18, we must have $V_\phi = V^*$ and $\pi_\theta = \pi^*$. Theorem 3 follows as a special case when the environment is deterministic. $\qquad\square$