[Reviews · NeurIPS 2017]

Reviewer 1



SUMMARY: The paper considers entropy regularized discounted Markov Decision Process (MDP), and shows the relation between the optimal value, action-value, and policy. Moreover, it shows that the optimal value function and policy satisfy a temporal consistency in the form of Bellman-like equation (Theorem 1), which can also be extended to its n-step version (Corollary 2). The paper introduces Path Consistent Learning by enforcing the temporal consistency, which is essentially a Bellman residual minimization procedure (Section 5). SUMMARY OF EVALUATION: Quality: Parts of the paper is sound (Section 3 and 4); parts of it is not (Section 5) Clarity: The paper is well-written. Originality: Some results seem to be novel, but similar ideas and analysis have been proposed/done before. Significance: It might become an important paper in the entropy regularized RL literature. EVALUATION: This is a well-written paper that has some interesting results regarding the relation between V, Q, and pi for the entropy regularized MDP setting. That being said, I have some concerns, which I hope the authors clarify. - Enforcing temporal consistency by minimizing its l2 error is minimizing the empirical Bellman error. Why do we need a new terminology (Path consistency) for this? - The empirical Bellman error, either in the standard form or in the new form in this paper, is a biased estimate of the true Bellman error. Refer to the discussion in Section 3 of Antos, Szepesvari, Munos, “Learning near-optimal policies with Bellman-residual minimization based fitted policy iteration and a single sample path,” Machine Learning Journal, 2008. Therefore, for stochastic dynamics, minimizing Equation (15) does not lead to the desired result. This is my most crucial comment, and I hope the authors provide an answer to it. - Claiming that this paper is bridging the gap between the value and policy based RL is beyond what has actually been shown. The paper shows the relation between value and policy for a certain type of reward function (which has the entropy of policy as an additional reward term). - On LL176-177, it is claimed that a new actor-critic paradigm is presented where the policy is not distinct from the values. Isn’t this exactly the same thing in the conventional value-based approach for which the optimal policy has a certain relationship with the optimal value function? In the standard setting, the greedy policy is optimal; here we have relationship of Equation (10). - It might be noted that relations similar to Equations (7), (8), (9), and (10) have been observed in the context of Maximum Entropy Inverse Optimal Control. See Corollary 2 and the discussion after that in the following paper: Ziebart, et al., “The Principle of Maximum Causal Entropy for Estimating Interacting Processes,” IEEE Transactions on Information Theory, 2013. Here the Bellman equations are very similar to what we have in this paper. Ziebart et al.’s framework is extended by the following paper to address problems with large state spaces: Huang, et al., “Approximate MaxEnt Inverse Optimal Control and its Application for Mental Simulation of Human Interactions,” AAAI, 2016. This latter paper concerns with the inverse optimal control (i.e., RL), but the approximate value iteration algorithms used for solving the IOC essentially use the same Bellman equations as we have here. The paper might want to compare the difference between these two approaches. - One class of methods that can bridge the gap between value and policy based RL algorithms is classification-based RL algorithms. The paper ignores these approaches in its discussion.

Reviewer 2



The paper derives a new insight for entropy-regularized RL: a relation between the optimal policy and value function that holds for any sequence of states along a trajectory. This relation, in the form of a consistency equation, is used to define a novel loss function for actor-critic algorithms, that holds in an off-policy setting as well as on-policy. The paper is clearly written and was a pleasure to read. Choosing to focus on deterministic transitions in the main text made it easier to focus on the main idea. While entropy based RL has been explored extensively, the insight in this paper is, to my knowledge, novel, and leads to a very elegant algorithmic idea. The off-policy actor critic proposed here is only slightly different from previous work such as A2C (as noted in the paper), but makes much more sense theoretically. Due to its principled construction, I believe that the PCL algorithm proposed here would become a standard RL algorithm in the future, and strongly suggest the acceptance of this paper. Comments: 120: Theorem 1: it would be interesting to follow up with the intuition of what happens when \tau -> 0 (line 102) for this result as well. I’d expect it to reduce to the hard Bellman backup, but for actions that are not optimal I cannot really see this. 74-77: Technically, the optimal policy is not necessarily unique. The uniqueness should be assumed for this paragraph to hold (this is not a strong assumption). I have read the author feedback.

Reviewer 3



The paper is well-written. I have two issues. First, I am not sure that NIPS the best place for it. You do mention neural networks as a possible application in the introduction, but that is about it. It seems to me, also given the theoretical nature of the publication. that JMLR or ICML would be better suited. In line 86 you describe two examples in which entropy regularisation has been shown to work. There are earlier examples: Information-Theoretic Neuro-Correlates Boost Evolution of Cognitive Systems by Jory Schossau, Christoph Adami and Arend Hintze Entropy 2016, 18(1), 6; doi:10.3390/e18010006 K. Zahedi, G. Martius, and N. Ay, “Linear combination of one-step predictive information with an external reward in an episodic policy gradient setting: a critical analysis,” Frontiers in psychology, vol. 4, iss. 801, 2013. Houthooft, Rein, et al. "Vime: Variational information maximizing exploration." NIPS 2016. It would be good, if the authors could expand on how their Equations 4 relates to other, previous approaches.